# Periodontal Disease Is Associated with Increased Vulnerability of Coronary Atheromatous Plaques in Patients Undergoing Coronary Computed Tomography Angiography—Results from the Atherodent Study

**DOI:** 10.3390/jcm10061290

**Published:** 2021-03-21

**Authors:** Ioana-Patricia Rodean, Luminița Lazăr, Vasile-Bogdan Halațiu, Carmen Biriș, Imre Benedek, Theodora Benedek

**Affiliations:** 1Clinic of Cardiology, Emergency Clinical County Hospital, 540136 Târgu Mureș, Romania; ioana_patricia91@yahoo.com (I.-P.R.); bhalatiu@yahoo.com (V.-B.H.); imrebenedek@yahoo.com (I.B.); theodora.benedek@gmail.com (T.B.); 2Center of Advanced Research in Multimodality Cardiac Imaging, CardioMed Medical Center, 540124 Târgu Mureș, Romania; 3Department of Cardiology, “George Emil Palade” University of Medicine, Pharmacy, Science and Technology, 540139 Târgu Mureș, Romania; 4Faculty of Dental Medicine, “George Emil Palade” University of Medicine, Pharmacy, Science and Technology, 540139 Târgu Mureș, Romania; biriscarmen74@yahoo.com

**Keywords:** atherosclerosis, computer tomography, periodontal disease, unstable angina, vulnerable plaques

## Abstract

The present study aimed to investigate the link between the severity of periodontal disease (PD), coronary calcifications and unstable plaque features in patients who underwent coronary computed tomography for unstable angina (UA). Fifty-two patients with UA, included in the ATHERODENT trial (NCT03395041), underwent computed tomographic coronary angiography (CCTA) and dental examination. Based on the median value of the periodontal index (PI), patients were assigned to the low periodontal index (LPI) group (PI < 22) and a high periodontal index (HPI) group (PI > 22). Patients with HPI had higher plaque volume (*p* = 0.013) and noncalcified plaque volume (*p* = 0.0003) at CCTA. In addition, the presence of vulnerability features in the atheromatous plaques was significantly correlated with PI (*p* = 0.001). Among periodontal indices, loss of gingival attachment (*p* = 0.009) and papillary bleeding index (*p* = 0.002) were strongly associated with high-risk plaques. PI significantly correlated with coronary calcium score (r = 0.45, *p* = 0.0008), but not with traditional markers of subclinical atherosclerosis. Overall, this subgroup analysis of the ATHERODENT study indicates that patients with advanced PD and UA present a higher amount of calcium in the coronary tree and have a more vulnerable phenotype of their culprit plaques.

## 1. Introduction

Acute coronary syndromes (ACS) represent major cardiovascular emergencies caused by acute occlusions of the coronary arteries, which may arise from atheromatous plaque disruption and/or thrombosis [1,2]. Even though current diagnostic and treatment techniques have been significantly improved, ACS remains the main cause of death in developing countries (31% of all deaths), while myocardial infarction (MI) leads the causes of mortality and morbidity worldwide, being responsible for 85% of cardiovascular deaths. Most cardiovascular diseases (CVD) are associated with atherosclerosis, a progressive chronic inflammatory process. Furthermore, inflammation represents the key mechanism of atherosclerosis since plaque progression and instability are triggered and favored by inflammation [3,4,5].

The PD is defined as a chronic inflammatory state produced by the imbalance of the normal oral microbiota, being characterized by the damage of the soft and hard structures that support the teeth as a result of repeated bacterial infections [1,4,6]. This inflammatory state leads to the local occurrence of a complex host inflammatory response, which is responsible for the progressive destruction of the supporting tooth tissue and leads to tooth loss [7,8,9,10].

Due to repeated bacteremia, pathogens are able to enter into the systemic circulation, favoring activation of the inflammatory cascade [6]. In addition, local inflammation causes changes in plaque morphology. Several studies demonstrated that CRP is directly involved in ACS, having a key role in intravascular inflammation and triggering plaque destabilization [11]. Macrophages infiltration within lipids, successive necrosis and apoptosis of these cells lead to the necrotic core formation and lipid-rich core, encapsulated by the surrounding fibrous tissue. These changes lead to the appearance of vulnerable plaques (VP), which can trigger an atherothrombotic event [7,9,12,13].

One of the most important predictors of atherosclerosis progression is the degree of calcification of the coronary arteries, which can be determined using coronary computed tomography angiography (CCTA). This technique can calculate the entire amount of coronary tree calcium, also known as the coronary calcium score (CACS) or Agatston score [14]. It is based on a semi-quantitative software able to identify and score calcium in the coronary tree and represents the sum of the calcium scores of all lesions, expressed in Hounsfield units (HU) [15]. The cardiovascular risk is classified according to Agatston score in low-risk if the score is between 0–99 HU, moderate risk between 100 and 399 HU, and high-risk, calcium score > 400 HU [14].

Rupture of an atheromatous plaque triggers a thrombotic cascade leading to a total or almost total occlusion of the coronary lumen [12]. The VP is coronary plaques with a high susceptibility to rupture, placing the patient at an increased risk for future major cardiac adverse events. A plaque at risk of rupture generally contains a large necrotic core surrounded by a thin fibrous cap called thin-cap fibroatheroma [12]. It was also observed that in more than 75% of the cases, fatal myocardial infarction is caused by the rupture of a coronary plaque [16]. Generally, a VP contains several markers of increased vulnerability that can be easily identified by CCTA. These markers include the presence of a low-density atheroma, spotty calcifications, positive remodeling, and a particular sign known as the “napkin-ring” sign, represented by a necrotic core, surrounded by a white area [10].

Inflammation plays a key role in the pathogenesis of atherosclerosis. PD is one of the most frequent inflammatory conditions associated with CVD, with the possible direct implications in the process of plaque vulnerabilization (Figure 1). Thus, the aim of our study was to evaluate the correlation between the severity of PD and the degree of vulnerability of atheromatous coronary plaques, as assessed by CCTA in patients with UA.

## 2. Materials and Methods

### 2.1. Study Population

ATHERODENT was a clinical trial (NCT03395041) conducted in two institutional sites: “George Emil Palade” the University of Medicine, Pharmacy, Science and Technology of Târgu Mureș, Romania, and the Center of Advanced Research in Multimodality Cardiac Imaging of the Cardio Med Medical Center, Târgu Mureș, Romania (protocol record number CM0117-ATD), aiming to compare the severity of coronary artery disease and plaque vulnerability in patients with PD versus those without PD. From the total cohort of the ATHERODENT trial, 52 patients presented UA and concomitant PD of various degrees. The present study is a sub-study of the ATHERODENT trial, including these 52 patients with UA in whom oral examination identified the presence of PD, enrolled between January and June 2018. This study was a case–control observational study aiming to investigate the association between coronary VP and severity of PD in patients with UA.

Diagnosis of UA was established according to the European Guideline for the management of acute coronary syndromes in patients presenting without persistent ST-segment elevation, as myocardial ischemia on minimal exertion or at rest without acute myocardial injury [17]. Thus, UA diagnosis was based on symptoms, electrocardiogram (ECG) criteria (no ST-elevation) and negative myocardial enzymes.

In all cases, personal and family medical history, as well as demographic data (age, gender, weight, height, body mass index -BMI), smoking status, abdominal circumference (AC), neck circumference and intima–media thickness (IMT), were recorded at baseline. Following an ECG and myocardial enzyme serum levels for ST-elevation (STEMI) and non-ST elevation myocardial infarction (non-STEMI) exclusion, all subjects underwent a CCTA examination for the assessment of the coronary tree and a complex dental examination. The exclusion criteria included STEMI, non-STEMI, contrast substance sensitivity, pregnancy, any malignancy, or comorbidity that can reduce life expectancy, renal failure, noncompliant patients, or less than 5 remaining teeth.

### 2.2. CCTA and Image Processing

The CCTA was performed using a 128-slice single-source computed tomography scanner—SOMATOM definition AS (Siemens AG, Healthcare Sector, Forchheim, Germany) with ECG gating. The following parameters were used for scanning: 100 kv tube voltage with 180 mAs/rotation tube current, gantry rotation time 330 msec, pitch adapted to heart rate and collimation 128 × 0.6. The total amount of contrast agent was individualized for each subject (according to body weight), and it was administrated in the antecubital vein at a 5 mL/s flow rate, using an 18-gauge venous catheter, followed by a saline flush. Images were acquired in two phases: before injecting the contrast substance in order to assess the calcium score, respectively, after the contrast agent was injected for visualization of the coronary tree. Image postprocessing and data analysis were performed using semi-automatic software provided by Siemens Syngo via Frontier through the Multimodality Research Platform from Cardio Med Medical Center.

### 2.3. Assessment of the Total Coronary Artery Calcium Score

Images obtained from the native thoracic scan were used for CACS assessment. This method is based on a semi-automatic software able to identify the presence of calcium in the coronary tree, which calculates the total CACS by summing the calcium from all coronary lesions. The result was expressed in HU [18].

### 2.4. Plaque Morphology

Plaque morphology was analyzed using semi-automatic software with the possibility of 3D reconstruction of the coronary arteries. In addition, the degree of stenosis, luminal and plaque volume, as well as composition (calcified, fibrotic or lipid-rich atheroma), were determined. For each lesion, the limits were marked, and internal and external contours were adjusted in order to reduce the errors. A complete characterization of the plaque composition in volume and percentage was obtained.

Vulnerability features were defined as (1) low attenuation plaque with a density < 50 HU, (2) positive remodeling (remodeling index > 1.1), (3) presence of spotty calcification, and (4) napkin ring sign. According to the vulnerable features, plaques were divided into two groups: high-risk plaques (with at least 2 vulnerability markers present) and low-risk plaques (1 vulnerability marker present or none) [19].

### 2.5. Dental Examination

A complex oral examination by an experienced periodontist was performed for periodontal status assessment prior to enrollment in the study. Oral examination investigated the presence of established features of PD, such as gingival index (GI), plaque index (PqI), calculus index (CI), furcation defects (FI)—described as the point at which two dental roots divided, papillary bleeding index (PBI), tooth mobility—assessed by holding between handles of the two instruments and moved back and forth the tooth, clinical attachment loss (CAL)—an irreversible sign of PD referring to the distance between the base of the pocket and a fixed point of the crown, and probing pocket depth (PPD)—defined as the distance between the base of the pocket and gingiva margins.

Quantification of the severity of the PD according to the periodontal index (PI) is exemplified in Table 1 [20,21,22].

### 2.6. Intima–Media Thickness Assessment

In order to assess the IMT, carotid ultrasound was performed by an experienced cardiologist using a Philips CX50 echocardiography device (Philips Medical Systems, Nederland BV) and 9 L ultrasound transducer. All images were acquired at the level of the carotid bifurcation, with the reference value established at 0.9 mm [23,24].

### 2.7. Study Groups

The study population consisted of 52 patients with UA, who underwent complex dental examination and CCTA, assigned into two groups based on the median value of the PI, which was 22. Group 1 enrolled 26 patients with PI ≤ 22, defined as LPI group, and group 2 included 26 patients with PI > 22, defined as HPI group.

### 2.8. Statistical Analysis

For statistical analysis, Graph Pad InStat 3.10 software (GraphPad Software Inc., San Diego, CA, USA) was used. Normality tests were performed for all data prior to statistical analysis. The two study groups were statistically compared in terms of demographics, periodontal status and presence of subclinical atherosclerosis using Mann–Whitney test (for non-normally distributed variables), Student’s *t*-test (for normally distributed variables) and Fisher’s test (for categorical variables). The obtained results were reported as number and percentages, mean +/− standard deviation (SD), relative risk (RR). The *p* value was set at 0.05 for statistical significance.

### 2.9. Ethical Consideration

The ATHERODENT study was ethically approved by the Ethics Committee of the Cardio Med Medical Center (approval no. 25/28.12.2017) and by the “George Emil Palade” University of Medicine, Pharmacy, Science and Technology of Târgu Mureș ethics committee (approval no. 351/13.12.2017). All study procedures were in line with the principles of the Declaration of Helsinki. Before any procedure, the subjects gave their permission by signing the informed consent. Moreover, all data were anonymized throughout the analysis.

## 3. Results

### 3.1. Basic Characteristics of the Study Population

The ATHERODENT study population included 52 patients (37 men (71,15%) and 15 women (28.85%)) presented for UA. The average age in the study group was 52.46 years old. The mean weight of the study population was 86.24 kg and the mean height 171.84 cm. Mean BMI was 29.09 kg/m^2^. Out of the 52 participants, 36 never smoked, and 14 were active smokers. Most of the smokers (30.76%) presented a higher PI but without statistical significance for the difference between the groups. In the LPI group, 15 participants were from urban areas, almost the same number from the HPI group (16 participants), and HPI was significantly correlated with the male gender (*p* = 0.008). Baseline characteristics of the study groups are presented in Table 2.

### 3.2. Periodontal Status and Total Coronary Calcium Score

CACS was significantly higher in the group with HPI compared with those with LPI (505.29 +/− 478.64 vs. 93.82 +/− 233.0, *p* = 0.0003—Figure 2A). Moreover, the value of PI was higher in patients with CACS higher than 100 HU (21 patients), compared to those with a lower CACS (31 patients)-(15.12 +/− 10.19 vs. 28.52 +/− 10.51, *p* < 0.0001—Figure 2B). Linear regression analysis of the relationship between PI and CACS demonstrated that the severity of the PD is directly correlated with the total CACS (*p* = 0.0008, r = 0.45—Figure 2C).

### 3.3. Periodontal Status and Plaque Vulnerability

Analysis of CCTA features of plaque vulnerability showed that in patients with VP features, such as low-density atheroma, positive remodeling, spotty calcification and napkin ring sign in the culprit lesions, the severity of the PI was higher compared to those with low-risk plaque (28.20 +/− 13.34 vs. 18.71 +/− 11.31, *p* = 0.02—Figure 3).

A subanalysis of the relationship between each PD index and plaque vulnerability revealed that from all PI indices, only PBI (4.5 +/− 3.06 vs. 2.04 +/− 1.96, *p* = 0.002) and CAL (3.6 +/− 2.91 vs. 1.66 +/− 1.8, *p* = 0.009) were significantly associated with the plaque vulnerability (Table 3).

In addition, a subanalysis of CCTA vulnerability markers showed that plaque volume (*p* = 0.013) calcified plaque volume (*p* = 0.01) and the noncalcified plaque volume (*p* = 0.003) showed significantly higher values in the HPI group compared with the LPI group (Table 4).

### 3.4. Periodontal Status and Subclinical Atherosclerosis

In respect to the correlation between PD and subclinical atherosclerosis markers, there were no significant differences between the LPI and HPI groups in terms of neck circumference (37.64 +/− 3.36 vs. 37.21 +/− 3.29, *p* = 0.6), AC (102.5 +/− 17.95 vs. 101.7 +/− 14.65, *p* = 0.8), mean IMT (0.94 +/− 0.14 vs. 0.88 +/− 0.26, *p* = 0.49), right IMT (0.92 +/− 0.24 vs. 0.87 +/− 0.21, *p* = 0.19) or left IMT (0.93 +/− 0.28 vs. 0.89 +/− 0.22, *p* = 0.32)—Figure 4.

## 4. Discussion

It is well-known that PD represents a chronic inflammatory disease that leads to increased cardiovascular risk and favors the development of an ACS; however, the association between PD, plaque vulnerability and atherosclerosis is not well understood yet. Thus, the ATHERODENT clinical trial aimed to investigate the link between PD severity and coronary plaque composition and morphology.

The CVD and PD are both non-communicable diseases (NCD) characterized by high prevalence worldwide [25]. Both of them are multifactorial diseases and share common risk factors, most of them linked with lifestyle [26]. The results from the ATHERODENT clinical trial showed that the association between PD and UA is more common in males, with an average age of 54 years, but no correlations regarding BMI, residency or smoker status were identified in this study. According to DeStefano et al., PD represents a significant risk factor for CVD independent of age, gender, residency, BMI, or comorbidities [27]. Similar findings were obtained in a study performed on 921 men without CVD at baseline. After 18 years follow-up period, they found that patients with PD are prone to CVD or stroke independent of any other cardiovascular risk factors, demographic data, BMI or smoker status (OR = 1.5 for total CVD, OR = 1.9 for fatal CVD) [28]. Another study conducted in Spain identified a higher prevalence of CVD among patients with PD, and this was almost double in the male gender. Furthermore, for both males and females, the risk was higher in an older population with poor socioeconomic status, diabetic or current smokers [29]. Opposite from these results, the study conducted by Pires et al. [30] proved that the risk of CVD is higher in obese patients with concomitant PD compared to those obese but without PD. Smoking and overweight are well-known to be risk factors for both CVD and PD. However, the results of our study showed no significant correlation between obesity or smoker status, PD severity and UA. This could be explained by the low number of patients enrolled in our study, while prevalence studies are usually based on large cohorts of subjects.

Another important finding of the ATHERODENT trial was the association between CACS and PD. It is well-known that CACS presents a high negative predictive value for CVD, and its value is strongly correlated with the disease severity. Thus, a CACS > 100 HU is associated with an increased risk of developing an ACS and higher mortality [31]. The results of the ATHERODENT trial show that the presence of PD may be associated with the progression of atherosclerosis, as indicated by a higher CACS. According to the author’s knowledge, this is the first study showing a relationship between CCTA markers of plaque vulnerability and severity of PD. In addition, patients with severe forms of PD who presented concomitant UA exhibited higher values of CACS [32].

A VP was typically defined as a plaque with a larger volume, mainly noncalcified and with a larger lipid-rich volume [31]. Our study showed that patients with UA and high PI present more frequent culprit plaques associated with an unstable phenotype. Furthermore, a positive correlation was observed between PD and the degree of plaque vulnerability. Analysis of plaque composition demonstrated that patients with severe forms of PD frequently present plaques with larger volumes, especially noncalcified volume and lipid-rich ones, compared to those with mild forms of PD or healthy ones. However, starting from this premise, in an IVUS study conducted by De Boer et al., no correlation between the presence of antibodies against PD and plaque vulnerability was identified [33]. Moreover, from all PI indices, PBI and CAL were more frequently identified in patients with the vulnerable phenotype of their culprit lesions. A recent study, conducted by Wojtkowska [34] and collaborators, showed similar results. Plaque accumulation and PBI were the most common associated with ACS (*p* < 0.001 and *p* = 0.001). In addition, a link between PD and carotid plaque instability was described. The CAL and PBI were significantly correlated with plaque instability (*p* = 0.002 and *p* = 0.009) [35]. A study characterizing the inflammatory link between PD and CVD, performed on 71 patients, indicated that PBI and PI are important predictors of ACS, concluding that patients with ACS present worse oral health status. Moreover, the severity of the PD was associated with the risk of developing an ACS, probably via an inflammatory pathway [36].

The association between subclinical atherosclerosis and PD in patients with UA was also investigated in our study. Recent data showed a significant correlation between PI and calcified carotid atheroma (CCA) (*p* = 0.02). Moreover, patients with CCA and PD were more prone to develop an ACS (*p* = 0.001) [37]. In addition, it was described that subclinical atherosclerosis, expressed by an increased IMT in patients with PD, is more frequent in men aged >45 years old (*p* < 0.05). Additionally, CAL was significantly correlated with CCA [38]. Pussinen et al. demonstrated in a study conducted on 755 participants that childhood infections, including PD, were correlated with subclinical atherosclerosis (increased IMT) in adulthood. The findings were more expressed in the male gender compared to females [39].

The results obtained in the present study did not reveal a statistical correlation between IMT and PD in patients with UA. However, a slight predominance of higher IMT was found in patients with severe forms of PD. An explanation for these findings may rely on the accelerated atherosclerotic process found in patients with UA. Hence, even non-obstructive plaques but with vulnerability features present may lead to the appearance of ACS. Similar findings were described in a study conducted by Bell et al., who did not identify any correlation between PD and subclinical atherosclerosis [40].

Further, in order to assess the presence of markers traditionally associated with subclinical atherosclerosis, neck circumference and AC were assessed in our study, but no correlation between these parameters and PD in patients with UA was identified. However, in a study performed on 950 patients, it was demonstrated that AC is significantly correlated with PD (*p* < 0.05) [41]. A study conducted by Saxlin and colleagues [42] concluded that AC is associated with severe forms of PD [43]. Moreover, in a recent study, it was found that AC is directly correlated with PPD, but only in non-diabetic patients, non-smoker subjects aged between 30 and 49 years old [34]. Similar results were also obtained in a recent study where a positive correlation between AC and PD was found. Moreover, it was observed that the female gender with PD presented an increased AC (*p* < 0.001) [44].

This study indicates a link between PD, the severity of atherosclerosis and plaque vulnerability. However, the biological plausibility, in this case, the cause–effect relationship between PD and atherosclerosis, remains uncertain. Since inflammation plays a pivotal role in the determinism of both diseases (PD and atherosclerosis), the direct contribution of the PD to atherosclerosis progression, independent of the inflammatory process, remains speculative. However, this study indicates that patients with PD and symptoms of heart disease should undergo a careful cardiologic examination since they may have additional cardiovascular risk resulting from the increased systemic inflammation triggered by the PD.

### Study Limitation

The main study limitation of this trial is related to the small number of patients with UA and concomitant PD included in the study trial. The main reason for the small number of patients was the fact that only patients with unstable angina who underwent CCTA as a first-line diagnosis were included in the study. In clinical practice, a large number of patients with UA still receive invasive angiography in an emergency instead of CCTA as a first-line diagnosis. This significantly limited the proportion of patients eligible for this study since those with UA receiving cardiac catheterization and stenting in an emergency prior to CCTA had to be excluded from the study because a coronary plaque already stented is no longer available for CCTA analysis.

Due to the low number of cases included in the study, the rate of clinical events was not analyzed. Since the patients were treated after CCTA (revascularized or with medical therapy), the estimated number of clinical events was too low to reach a statistical significance between the groups.

A potential bias of the study may result from the patient selection since this is a subgroup analysis of the ATHERODENT clinical study. Only patients with PD and UA were included in this subanalysis, which represents a high-risk group. A prospective study design would allow assessment of the real influence between the studied variables, but this is the subject of the larger ATHERODENT trial that also includes follow-up data.

## 5. Conclusions

This subgroup analysis of the ATHERODENT clinical study indicates that patients with advanced PD and UA present a higher amount of calcium in the coronary tree and have a more vulnerable phenotype of their culprit plaques. Overall, the results of this study indicate an association between severity of PD, the severity of atherosclerosis and coronary plaque vulnerability in patients with UA, probably having systemic inflammation as a common substrate.

## Figures and Tables

**Figure 1 jcm-10-01290-f001:**
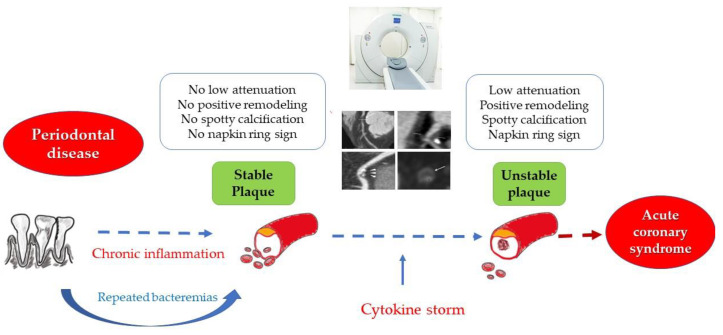
The link between periodontal disease, chronic inflammation and plaque vulnerability.

**Figure 2 jcm-10-01290-f002:**
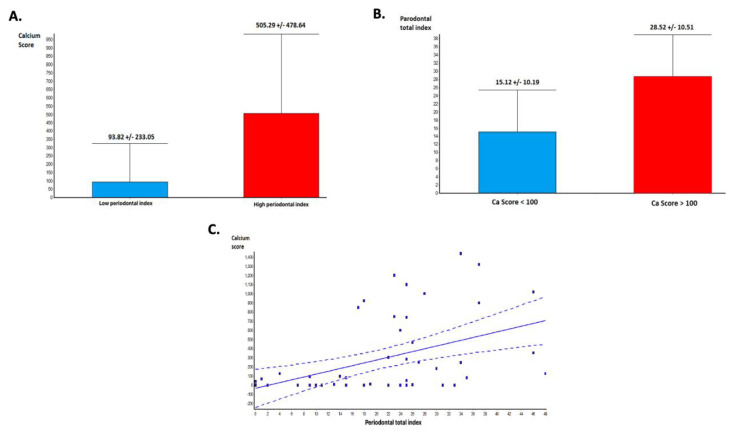
Periodontal disease and coronary calcium score: (**A**) Calcium score and periodontal index (PI). Calcium score is significantly higher in patients with advanced periodontal disease (PD). (**B**) PI is significantly higher in patients with high calcium scores. (**C**) Linear regression analysis demonstrating a significant association between total calcium score and periodontal index.

**Figure 3 jcm-10-01290-f003:**
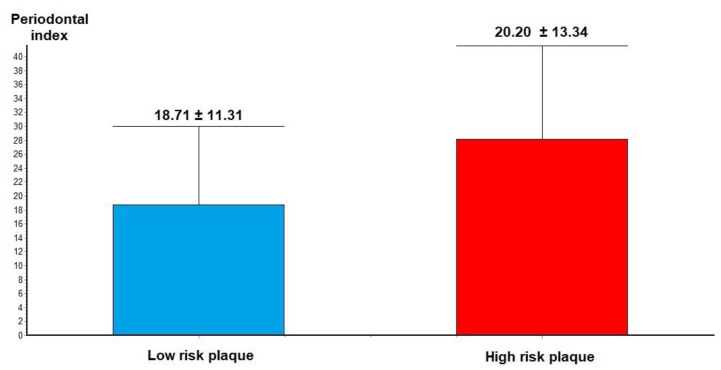
Periodontal index in low-risk plaque vs. high-risk plaque.

**Figure 4 jcm-10-01290-f004:**
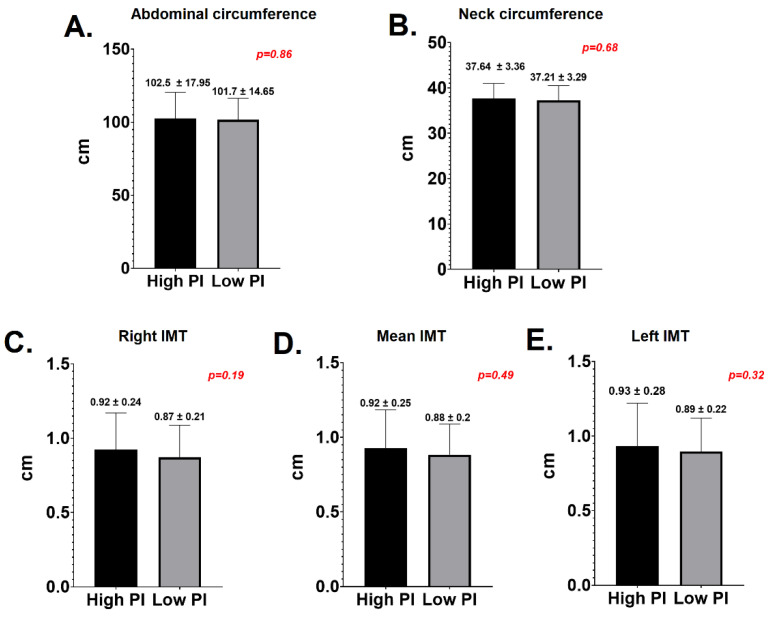
Periodontal diseases and markers of subclinical atherosclerosis. (**A**) Abdominal circumference; (**B**) neck circumference; (**C**) right intima media thickness; (**D**) mean intima media thickness; (**E**) left intima media thickness.

**Table 1 jcm-10-01290-t001:** Periodontal indices quantification.

Periodontal Indices	0	1	2	3	4
GI	Normal, healthy gum	Mild inflammation	Moderate inflammation	Advanced inflammation	-
PqI	No plaque	Plaque < 1/3 of the tooth surface	Plaque between 1/3–2/3 of the tooth surface	Plaque > 2/3 of the tooth surface	-
CI	No calculus	Calculus < 1/3 of the tooth surface	Calculus between 1/3–2/3 of the tooth surface	Calculus > 2/3 of the tooth surface	-
FI	The furcation probe cannot enter the furcation area	The probe is able to partially enter the furcation, extending less than 1/3 of the width of the tooth	The probe extends more than ½ of the width of the tooth but does not pass completely through the furcation	The probe passes completely through the furcation	-
PBI (Muehlemann index)	No bleeding	Bleeding isolated visible one point	Multiple bleeding points	Red line in the margin	Heavy bleeding or profuse
Tooth mobility	No apparent mobility	Horizontal, bucco–oral movement < 1 mm	Horizontal, bucco–oral movement >1 mm	Horizontal, bucco-oral movement >1 mm, also in the axial direction	-
CAL	No attachment loss	Mild, between 1 and 2 mm	Moderates between 3 and 4 mm	Severe, >5 mm	-
PPD	Distance between 1 and 3 mm	Distance between 3 and 5 mm	Distance between 5 and 7 mm	Distance > 7 mm	-

The PI was defined as the sum of all 8 examined parameters, and it was calculated for the tooth’s 1.6, 2.1, 2.4, 3.6, 4.1, 4.4. According to the median PI, set by 22, the periodontal status was divided into LPI (healthy gums or gingivitis) and HPI (periodontitis and severe periodontitis).

**Table 2 jcm-10-01290-t002:** Demographic and risk factors data of the study groups.

Variables	Group 1—LPI		Group 2—HPI		*p* VALU
Mean age (y)					
	50.27		54.65		0.2

Mean Weight (kg)	87.5		85.04		0.6

Mean height (cm)					
	173.90		169.5		0.1

Mean BMI (kg/m^2^)	28.75		29.49		0.6

Gender distribution N (%)	Male	Female	Male	Female	0.008
	17 (65.38%)	9 (34.61%)	20 (76.92%)	6 (23.08%)	
Residency N (%)	Urban	Rural	Urban	Rural	0.6
	15 (57.69%)	11 (42.31%)	16 (61.53%)	9 (38.47%)	
Smoker status N (%)	Smoker	Non-smoker	Smoker	Non-smoker	0.2
	6 (23.07%)	20 (76.93%)	8 (30.76%)	16 (69.24%)	

**Table 3 jcm-10-01290-t003:** Periodontal indices in low-risk plaque vs. high-risk plaque.

Periodontal Indices	Low-Risk Plaque	High-Risk Plaque	*p* Value
GI	3.5 +/− 2.21	5 +/− 2.66	0.07
PqI	5.14 +/− 4.02	7.5 +/− 3.97	0.1
CI	2.74 +/− 2.14	2.6 +/− 1.83	0.7
FI	0.21 +/− 0.56	0.4 +/− 1.26	0.4
PBI	2.048 +/− 1.96	4.5 +/− 3.064	0.002
Mobility	2.83 +/− 2.56	3.2 +/− 2.57	0.6
CAL	1.66 +/− 1.8	3.6 +/− 2.91	0.009
PPD	0.83 +/− 1.2	1.4 +/− 1.35	0.2

**Table 4 jcm-10-01290-t004:** Plaque vulnerability markers in low versus high-risk plaques.

	**Low-Risk Plaque**	**High-Risk Plaque**	***p* Value**
Plaque volume	143.88 +/− 217.08	320.89 +/− 329.27	0.01
Calcified volume	109.08 +/− 157.46	270 +/− 297.41	0.01
Noncalcified plaque volume	34.85 +/− 89.77	55.02 +/− 65.75	0.003
Lipid-rich plaque volume	15.77 +/− 53.37	10.29 +/− 11.87	0.05

## Data Availability

The data presented in this study are available on request from the corresponding author. The data are not publicly available due to personal protection.

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
