# Peer review of "Periodontal Disease Is Associated with Increased Vulnerability of Coronary Atheromatous Plaques in Patients Undergoing Coronary Computed Tomography Angiography—Results from the Atherodent Study"

_jcm, 2021, doi:10.3390/jcm10061290_

Round 1
Reviewer 1 Report
The paper entitled «Periodontal disease is associated with increased vulnerability 2 of coronary atheromatous plaques in patients undergoing Coro-3 nary Computed Tomography Angiography - results from the 4 ATHERODENT study” reports interesting data on periodontal disease as a risk factor for CVD in patients admitted for unstable angina.
The paper is easy to follow and well presented.
The methods are clearly presented, the step-by-step approach is easy to follow.
Main concerns
- The introduction appears as a mini-review on the topic. This is well-written and informative but really too long as an introduction to a research paper. It should be shortened. Similarly the discussion should present shortly the main findings and the main points for discussion.
- Be careful the NCT trial is not always reported consistently in the manuscript. The good one appears to be NCT03395041.
- In the registered trail, number of patients should be 100. This contrasts with the 52 patients finally included. This is a blocking point, to my view. Please explain
- Similarly, the limitation section should be developed. What were the hypotheses leading to this small number of patients? PD is very frequent, UA is frequent, recruitment should have been easy. Please comment.
- Are other endpoints available, such as troponin AUC, or clinical events?
Minor concerns
- Please check the units. For instance regarding BMI Kg/cm2 is uncorrect.
- As regards methods, the authors should indicate references to support their descriptions (scan for instance)
Reviewer 2 Report
The present study aims to analyze the correlation between periodontal disease and atherosclerosis. This study topic is really interesting due to the prevalence and severity of cardiovascular disease. Furthermore, the association between some systemic pathology and periodontal disease is a current research topic.
The authors have adequately worked on the study, so the manuscript deserves to be published. However, the following questions could improve the job.
Title
The title is correct and descriptive of the work.
Introduction
The introduction is appropriate according to the topic addressed. It is descriptive of the periodontal and cardiovascular disease, highlighting the severity of the patient's health. In addition, the possible link between both pathologies is analyzed.
Material and Methods
The methodology employed in the clinical trial is well detailed, specifying characteristics of the population, aspects related to both cardiovascular (CCTA and image processing, coronary artery calcium score or intima media thickness assessment) and periodontal disease, and statistical analysis.
Results
The results of the study, together with the analyzed statistics, are presented correctly, including tables and figures to clarify the content for the reader.
In table 2, each study group should show the numerical p-value, although the distribution by gender must also be marked as significant. The same in table 3.
Discussion
The discussion shows an interesting analysis of the results of the study in comparison with other studies present in the literature. However, the authors confuse association with causality on numerous occasions. Some discussion should be analyzed again and rewritten taking into account this premise: not every association implies causality between both variables (see Bradford Hill
Other hand, biological plausability of the results obtained must be shown in the discussion.
Moreover, an analysis of the risk of bias of the study carried out would be interesting, as well as an assessment of a prospective study design that allows evaluating the real influence between both variables.
Round 2
Reviewer 1 Report
We would like the authors for their answers.